# Improving the Misalignment Tolerance of Wireless Power Transfer System for AUV with Solenoid-Dual Combined Planar Magnetic Coupler

**Haibing Wen [1,\*], Peng Wang [1], Jiayuan Li [1], Jiadong Yang [1], Kehan Zhang [2], Lei Yang [1], Yaopeng Zhao [1] and Xiangqian Tong [1]**

[1] School of Electrical Engineering, Xi'an University of Technology, Xi'an 710048, China; 13720589683@163.com (P.W.); lijiayuan1998@126.com (J.L.); yangjiadongyy@163.com (J.Y.); yanglei0930@xaut.edu.cn (L.Y.); ypzhao@xaut.edu.cn (Y.Z.); xqtong@xaut.edu.cn (X.T.)

[2] School of Marine Science and Technology, Northwestern Polytechnical University, Xi'an 710072, China; zhangkehan210@163.com

\* Correspondence: wenhaibing@xaut.edu.cn

**Abstract:** In order to solve the problem of power transmission efficiency reduction resulting from misalignment in the Wireless Power Transfer (WPT) system for Autonomous Underwater Vehicle (AUVs), a novel coupling structure with strong tolerance to misalignment is proposed. A solenoid coil is selected as the transmitting coil, and the receiving coil is composed of dual combined planar coils. The WPT system can still maintain stable output under uncertain axial misalignments for AUVs. The magnetic field distribution of the proposed magnetic coupling structure is analyzed theoretically, and the distance between the coils in the dual combined planar receiving coil is optimized. The theoretical analysis shows that the proposed solenoid-dual combined planar coils coupling structure can effectively maintain a stable mutual inductance between the transmitting coil and receiving coil under different axial misalignments compared with solenoid-unipolar planar coil coupling structure. An S-S resonant compensated WPT experimental prototype is built to verify the output characteristics of the proposed magnetic coupling structure. Compared to the magnetic coupler with the unipolar planar coil, it is validated by experiment that the proposed magnetic coupler substantially enhances the stability of power transmission efficiency and output power when axial misalignment occurs. The power transmission efficiency decreases by 6.74% when axial misalignment increases from 0 to 40 mm in saltwater. Meanwhile, the variation of output power is less than 4.15%.

**Keywords:** wireless power transfer (WPT); autonomous underwater vehicle (AUV); misalignment tolerance; power transmission efficiency; output power

## 1. Introduction

AUVs have great potential application prospects in the exploitation of marine resources [1,2]. The energy issue is the main factor restricting the long-range continuing operation of underwater vehicles in the ocean. At present, two main methods for supplying electric energy to underwater vehicles are employed: the first is to use mechanical equipment to salvage the underwater vehicle and manually replace the battery; the other method is to use a wet-mate charging connector on an underwater power supply platform to charge the AUV [3]. The former method has a series of shortcomings such as high labor cost, low degree of automation, and poor concealment in salvage operations, and a lot of electrical energy is wasted on the round-trip journey of the underwater vehicles, resulting in low energy utilization rate. The latter method requires precise docking and complex insertion and extraction operations, which cause an issue with wearing; additionally, the price of wet-mate connectors is rather high. Magnetic coupling WPT based on electromagnetic induction is a new type of power transmission method that has been emerging and

developing over the last two decades, and which possesses the advantages of reliability and safety [4,5]. In WPT systems, the physical connections between the power sending side and the power receiving side are eliminated, effectively avoiding safety hazards such as the leakage, sparks, and carbon deposition characterizing traditional contact power transmission methods, thus enhancing the freedom and safety of electrical energy transmission. WPT technology has gradually been maturing over the last decade, and it has been extensively adopted in fields such as electric vehicles, biomedical implants, industrial manufacturing, and smart homes [6–8]. The use of WPT technology in the marine environment for power supply of underwater vehicles does not require direct electrical connection between the AUV and the submarine base station, thus avoiding the complex operation process, strict positioning accuracy requirements, and safety hazards that characterize wet-mate charging connectors, giving this method broad application prospects [9–11].

The electromagnetic coupling mechanism is a crucial element of WPT systems, and the magnetic coupling mechanism determines the vital characteristics of the system, such as power transmission efficiency, output power and transmission distance [12–14]. Due to the unique structure of AUVs, it is also necessary to consider the docking process between the underwater charging base station and AUVs, as well as the position deviation caused by seawater when designing a WPT system for AUVs. Researchers have designed various coupler structures for AUV WPT systems. Yan et al. [15] developed a coupler with a curly coil structure, and both unipolar and bipolar curly coils were studied. Compared with the unipolar structure, the bipolar curly coil was able to mitigate the electromagnetic interference of the AUV. A 1000 W experimental platform was established, and the power transmission efficiency reached 95%. In [16], a magnetic coupler with a coaxial solenoid coil structure suitable for AUV was proposed with a maximum output power of 300 W and a power transmission efficiency of about 63~77%. The WPT system was improved through the introduction of an LCC-S topology; the output power increased to 882 W, and the efficiency was enhanced to 81.3% [17]. Zeng et al. [18] developed a hybrid transmitter structure consisting of a conical spiral coil and a planar spiral coil. The receiver was a solenoid coil, which was wound around the AUV. The magnetic coupler structure could form a circular uniform magnetic field; thus greatly improving the misalignment tolerance of WPT system. A novel three-phase coil structure composed of three pairs of transmitters and receivers was presented in [19]; the magnetic flux density produced by the coupling coil structure was centralized within the magnetic coupler, thus avoiding the electronic components in the AUV being subjected to electromagnetic interference. Yan et al. [20] designed a magnetic coupler for an AUV wireless charging system, where the coupling structure consisted of a transmitting coil and two reverse-wound double-layer receiving coils; it showed great angular tolerance to rotational offsets. In [21], a coupling mechanism was proposed in which the transmitter was composed of circularly distributed multiple coils connected in series, and the receiver consisted of two vertical cross coils; the proposed coupling structure had good tolerance to rotational misalignments. In order to reduce the eddy current loss, Zhang et al. [22] put forward a new magnetic structure for AUVs, consisting of two transmitting coils and a receiving coil; the eddy current loss decreased by 50% compared to the conventional one-transmitter-to-one-receiver structure. However, two main challenges have restricted the applications of the above coupling structures in WPT systems for AUVs. On one hand, some coupling structures are incompatible with AUVs' profiles, resulting in inevitable alterations, which may cause harmful effects to the hydrodynamic characteristics and reduce the compressive strength of AUVs. On the other hand, some uncertain misalignments can easily be caused by docking errors or seawater flow vibration in the marine environment; it is difficult to design a magnetic coupler that simultaneously possesses high tolerance to radial, axial, and rotational misalignments.

Yao et al. [23] proposed a circular-cylindrical magnetic coupling structure consisting of a planar spiral transmitting coil and a solenoid receiving coil. It was verified that the leakage magnetic flux at both ends of the solenoid coil was small, and the coupling structure was suitable for mid-range power transmission scenarios. Cai et al. [24] developed

a magnetic structure consisting of a combined arc-shaped bipolar transmitting coil and a compact dipole receiving coil for AUV wireless charging systems. Stable magnetic coupling against three-dimension misalignments was established by using horizontal magnetic flux, effectively improving the WPT system's misalignment tolerance against rotational, axial and radial offsets. It was verified by experiment that the maximum output power reached 1.05 kW, and the dc-dc efficiency was 95.1%.

Under the dynamic underwater environment, impacts on wireless AUV charging systems caused by seawater flow are inevitable; in order to address the issue of reduced power transmission efficiency resulting from misalignments in underwater WPT systems, inspired by the combined coils idea and circular-cylindrical magnetic coupler mentioned in [23], this paper proposes a novel magnetic coupling structure composed of a solenoid coil as the transmitting coil and dual combined planar coils as the receiving coil. The proposed magnetic coupling structure is shown in Figure 1. The solenoid transmitting coil is wound in the submarine base station, and the coils in the dual combined planar receiving coil are installed in the AUV. This novel magnetic coupling structure enables a stable power transmission efficiency and output power to be achieved in a wireless charging system for AUVs, thus enhancing the misalignment tolerance of the WPT system.

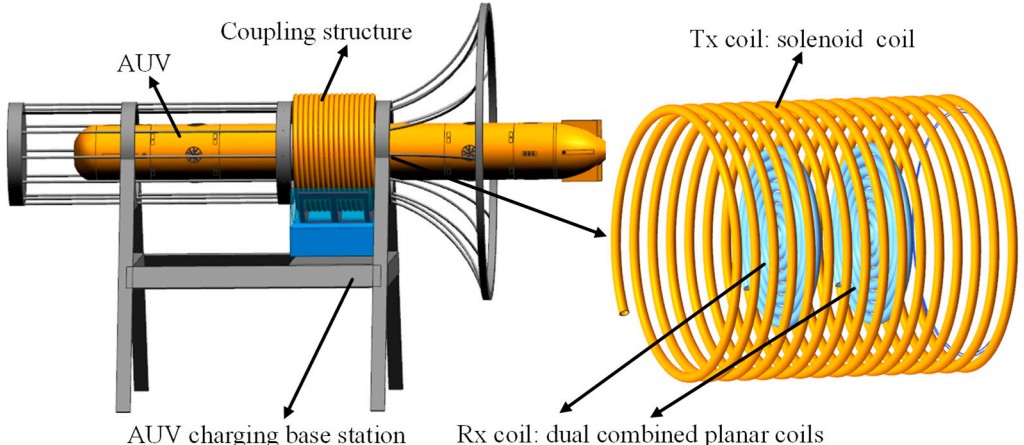

**Figure 1.** The proposed coupling structure.

This paper is organized as follows. Section 2 presents the theoretical analysis of the equivalent circuit model. The magnetic field distribution of the coupling structure is given in Section 3, and the parameters of the coupling structure are optimized. An experimental system is built to confirm the effectiveness of the presented magnetic coupler in Section 4. Section 5 discusses the advantages of and main issues with this novel coupling structure in WPT systems for AUVs. Finally, the conclusions are drawn in Section 6.

## 2. Circuit Model Analysis

Figure 2 presents the equivalent circuit of the WPT system with series–series compensation. On the primary side, a full-bridge inverter consisting of four MOSFETs ($S_1$-$S_4$) is utilized to produce ac power through the input dc voltage $U_{dc}$. The magnetic coupler is composed of a transmitting coil $L_1$ and receiving coils $L_{S1}$ and $L_{S2}$. $M_1$ and $M_2$ represent the coupling between the transmitting coil and the dual receiving coil, respectively, and $M_3$ represents the coupling between the dual receiving coils $L_{S1}$ and $L_{S2}$. Owing to the electromagnetic induction of the coupling structure, the electrical power can be transmitted wirelessly from the power sending side to the receiving side. The compensation capacitors $C_P$ and $C_S$ are chosen to achieve a resonant state with primary inductance and secondary inductance, respectively. On the secondary side, four DIODEs ($D_1$-$D_4$) make up the rectifier, and it is employed to convert ac power to dc voltage.

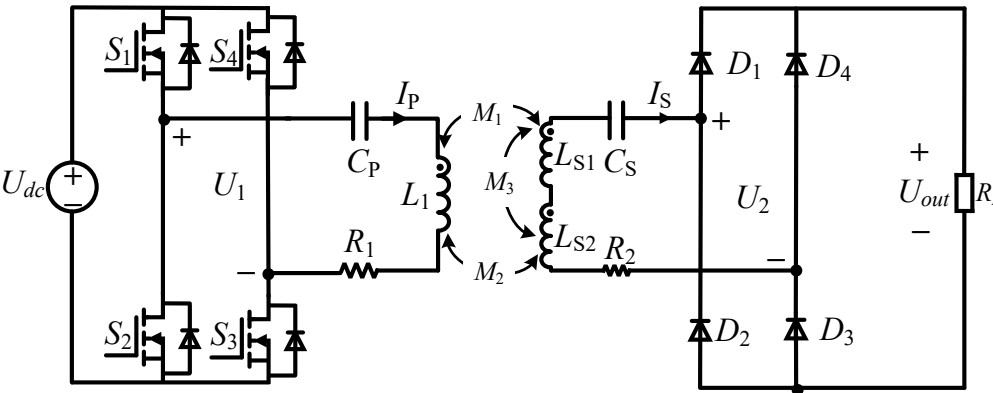

**Figure 2.** The equivalent circuit model of a WPT system with the proposed coupling structure.

The total mutual inductance between the transmitting coil and the dual receiving coil remains at a constant value when axial misalignment occurs; on the secondary side, the dual combined receiving coil can be equivalent to a coil with a self-inductance of $L_{S1} + L_{S2} - 2M_3$. To simplify the analysis, a new coil with a self-inductance of $L_2 = L_{S1} + L_{S2} - 2M_3$ is introduced to serve as an equivalent replacement for the dual combined planar coils, and Figure 3 shows the simplified equivalent circuit.

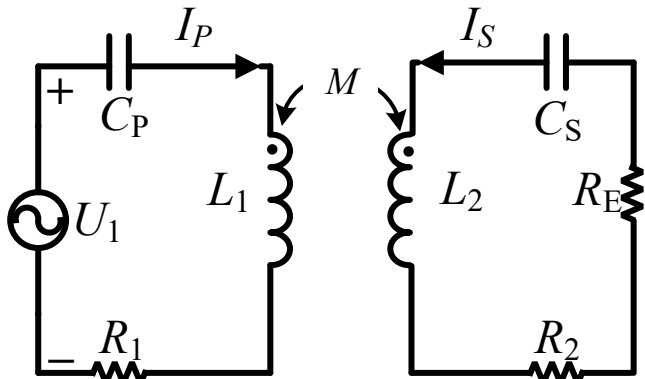

**Figure 3.** Simplified equivalent circuit of the WPT system.

According to Kirchhoff's voltage law, the expressions can be acquired as follows.

$$\begin{cases} I_P \left( \frac{1}{j\omega C_p} + j\omega L_1 + R_1 \right) + I_S j\omega M = U_1 \\ I_P j\omega M + I_S \left( j\omega L_2 + \frac{1}{j\omega C_S} + R_2 + R_E \right) = 0 \end{cases} \tag{1}$$

where $I_P$ and $I_S$ are the current flow through the transmitting coil and receiving coil, $\omega$ is the angular frequency of the WPT system, $U_1$ is the input voltage, and $R_1$ and $R_2$ are the internal resistance of the transmitting coil and receiving coil, respectively. $R_E = 8R_L/\pi^2$ is the equivalent load resistance.

When the WPT system is in the resonant state, the following equation can be obtained.

$$\begin{cases} \frac{1}{j\omega C_p} + j\omega L_1 = 0 \\ j\omega L_2 + \frac{1}{j\omega C_S} = 0 \end{cases} \tag{2}$$

Then, Equation (1) can be simplified to

$$\begin{cases} I_P R_1 + I_S j\omega M = U_1 \\ I_P j\omega M + I_S(R_2 + R_E) = 0 \end{cases} \tag{3}$$

$I_P$ and $I_S$ can be obtained from Equation (3) as follows:

$$\begin{cases} I_P = \frac{U_1(R_2+R_E)}{\omega^2 M^2 + R_1(R_2+R_E)} \\ I_S = \frac{U_1 j\omega M}{-\omega^2 M^2 - R_1(R_2+R_E)} \end{cases} \tag{4}$$

Actually, in the whole WPT system, compared with the equivalent load resistance $R_E$, the internal resistance of the transmitting coil and receiving coil $R_1$ and $R_2$ are negligible. Thus, the current of the primary side $I_P$ and the current of the secondary side $I_S$ can be expressed as

$$\begin{cases} I_P = \frac{U_1 R_E}{\omega^2 M^2} \\ I_S = \frac{U_1}{\omega M} \end{cases} \tag{5}$$

According to Equation (5), the secondary current $I_S$, namely the output current, is independent of load resistance, and it is related to the mutual inductance between the transmitting coil and the receiving coil. The output of the WPT system is characterized by constant current.

The output power of the WPT system $P_{out}$ and the power transmission efficiency $\eta$ can be calculated using (6) and (7), respectively.

$$P_{out} = I_S{}^2 R = \frac{U_1{}^2 \omega^2 M^2}{[-\omega^2 M^2 - R_1(R_2 + R_E)]^2} R_E \tag{6}$$

$$\eta = \frac{P_{out}}{P_{in}} = \frac{R_E(R_2 + R_E)(\omega^2 M^2)}{\omega^2 M^2 + R_1(R_2 + R_E)} \tag{7}$$

It can be seen from Equations (6) and (7) that both the output power $P_{out}$ and power transmission efficiency $\eta$ of the WPT system are influenced by the mutual inductance of the magnetic coupler. If the variation in mutual inductance could be reduced as the axial misalignment emerges, the stability of the WPT system's output characteristics would be enhanced.

## 3. Coupling Structure Design

### 3.1. Theoretical Analysis of Magnetic Field Distribution

A solenoid coil is composed of multiple single-turn coils with the same radius; thus, the magnetic field distribution of a solenoid coil is the vector sum of the magnetic field generated by the multiple single-turn coils. A cylindrical coil with the coordinates $O$-$\rho\phi z$ is built, as shown in Figure 4. The z-axis is perpendicular to the coil and passes through the center of the coil. When $t = 0$, the current direction in the coil is a right-handed spiral in the positive direction of the z-axis, and the expression of the current is $i(t) = \sqrt{2}I\cos(\omega t)$. The radius of the coil is $a$, and the coil is located in the plane $\Gamma$ ($z = h$); it is specified that the $z < h$ region is area 1, and the $z > h$ region is area 2. The seawater can be regarded as a linearly uniform infinite medium.

The electric field intensity at a point $Q(\rho, \phi, z)$ can be derived using the Maxwell equations [25].

$$\boldsymbol{E}(\rho, \phi, z) = -\frac{j\omega\mu a I}{2} \int_0^\infty \frac{\lambda}{u} J_1(\lambda a) J_1(\lambda\rho) e^{-u|z-h|} d\lambda \boldsymbol{e}_\phi \tag{8}$$

where $\omega$ is the angular frequency of the current, $\mu$ is the permeability of seawater, $\lambda$ is the separation constant, and $J_1(x)$ is the first-species first-order Bessel function. $u$ can be expressed as

$$u = \sqrt{\lambda^2 + j\omega\mu(\sigma + j\omega\varepsilon)} \tag{9}$$

where $\sigma$ is the conductivity of seawater and $\varepsilon$ is the permittivity of seawater.

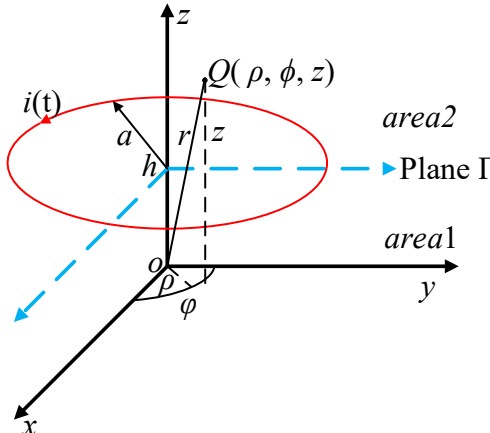

**Figure 4.** Single-turn coil with cylindrical coordinates.

The relationship between the magnetic field intensity $H$ at a point $Q(\rho, \phi, z)$ and the electric field intensity $E$ satisfies the following equation:

$$H = -\frac{1}{j\omega\mu}\nabla \times \left(E_\phi e_\phi\right) \tag{10}$$

Therefore, by substituting Equation (8) into Equation (10), the magnetic field intensity at a point $Q(\rho, \phi, z)$ can be derived as

$$H(\rho, \phi, z) = \frac{aI}{2}\int_0^\infty [\mathrm{sgn}(z-h)J_1(\lambda\rho)e_\rho + \frac{\lambda}{u}J_0(\lambda\rho)e_z]\lambda J_1(\lambda a)e^{-u|z-h|}d\lambda \tag{11}$$

where

$$\mathrm{sgn}(z-h) = \begin{cases} 1, & z > h \\ 0, & z = h \\ -1, & z < h \end{cases}$$

$J_0(x)$ is the first-species zero-order Bessel function.

A solenoid coil is composed of multiple single-turn coils, as shown in Figure 5. The number of turns is $N$, each single-turn coil is labeled as the 1st, 2nd, ..., and $N$th coil, respectively. In Figure 5, the blue circle represents the initial position of single-turn coil, and the red circles stand for the new positions after moving along $z$-axis. Therefore, the $N$th single-turn coil can be regarded as translating the 1st single-turn coil along the positive $z$-axis direction $(N-1)$ times, with a spacing of $s$ between each coil.

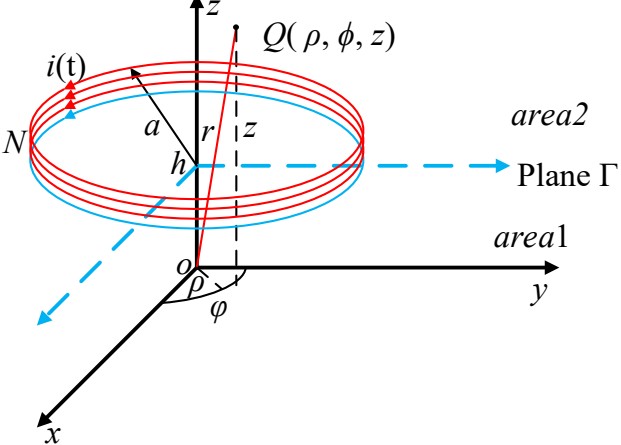

**Figure 5.** A solenoid coil in cylindrical coordinates.

The analytical expression of the magnetic field intensity at a point $Q(\rho, \phi, z)$ produced by a single-turn coil after translating the 1st single-turn coil along the positive $z$-axis direction with a spacing of $s$ can be derived as follows:

$$\boldsymbol{H}(\rho, \phi, z) = \frac{aI}{2} \int_0^\infty \left[\text{sgn}(z - (h+s))J_1(\lambda\rho)\boldsymbol{e}_\rho + \frac{\lambda}{u}J_0(\lambda\rho)\boldsymbol{e}_z\right]\lambda J_1(\lambda a)e^{-u|z-(h+s)|}d\lambda \quad (12)$$

where

$$\text{sgn}(z - (h+s)) = \begin{cases} 1, & z > h + s \\ 0, & z = h + s \\ -1, & z < h + s \end{cases}$$

It can be concluded that the magnetic field intensity at a point $Q(\rho, \phi, z)$ generated by a single-turn coil after translating the 1st single-turn coil along the positive $z$-axis direction $(N-1)$ times can be obtained as follows:

$$\boldsymbol{H}(\rho, \phi, z) = \frac{aI}{2} \int_0^\infty \left[\text{sgn}(z - (h + (N-1)s))J_1(\lambda\rho)\boldsymbol{e}_\rho + \frac{\lambda}{u}J_0(\lambda\rho)\boldsymbol{e}_z\right]\lambda J_1(\lambda a)e^{-u|z-(h+(N-1)s)|}d\lambda \quad (13)$$

where

$$\text{sgn}(z - (h + (N-1)s)) = \begin{cases} 1, & z > h + (N-1)s \\ 0, & z = h + (N-1)s \\ -1, & z < h + (N-1)s \end{cases}$$

The magnetic field intensity at a point $Q(\rho, \phi, z)$ generated by the solenoid transmitting coil is the vector sum of magnetic field intensity induced by the 1st, 2nd, ..., and $N$th single-turn coils. The analytical expression of magnetic field intensity is calculated as follows:

$$\begin{aligned} \boldsymbol{H}(\rho, \phi, z) = \ & \frac{aI}{2} \int_0^\infty \lambda J_1(\lambda a) \Big\{ \left[\text{sgn}(z - h)J_1(\lambda\rho)\boldsymbol{e}_\rho + \frac{\lambda}{u}J_0(\lambda\rho)\boldsymbol{e}_z\right]e^{-u|z-h|} \\ & + \left[\text{sgn}(z - (h+s))J_1(\lambda\rho)\boldsymbol{e}_\rho + \frac{\lambda}{u}J_0(\lambda\rho)\boldsymbol{e}_z\right]e^{-u|z-(h+s)|} + \dots \\ & + \left[\text{sgn}(z - (h + (N-1)s))J_1(\lambda\rho)\boldsymbol{e}_\rho + \frac{\lambda}{u}J_0(\lambda\rho)\boldsymbol{e}_z\right]e^{-u|z-(h+(N-1)s)|} \Big\}d\lambda \end{aligned} \quad (14)$$

The magnetic induction intensity at a point $Q(\rho, \phi, z)$ can be obtained as follows:

$$\begin{aligned} \boldsymbol{B}(\rho, \phi, z) = \ & \frac{\mu aI}{2} \int_0^\infty \lambda J_1(\lambda a) \Big\{ \left[\text{sgn}(z - h)J_1(\lambda\rho)\boldsymbol{e}_\rho + \frac{\lambda}{u}J_0(\lambda\rho)\boldsymbol{e}_z\right]e^{-u|z-h|} \\ & + \left[\text{sgn}(z - (h+s))J_1(\lambda\rho)\boldsymbol{e}_\rho + \frac{\lambda}{u}J_0(\lambda\rho)\boldsymbol{e}_z\right]e^{-u|z-(h+s)|} + \dots \\ & + \left[\text{sgn}(z - (h + (N-1)s))J_1(\lambda\rho)\boldsymbol{e}_\rho + \frac{\lambda}{u}J_0(\lambda\rho)\boldsymbol{e}_z\right]e^{-u|z-(h+(N-1)s)|} \Big\}d\lambda \end{aligned} \quad (15)$$

### 3.2. Simulation Analysis and Parameter Optimization

In order to simulate the magnetic induction intensity distribution of the proposed coupling structure, a simulation model is built using the finite element simulation software COMSOL. The solenoid coil is selected as the transmitting coil, and the dual combined planar coils are used as the receiving coil. The parameters of the proposed magnetic coupler are listed in Table 1.

Figure 6 presents a sectional diagram of the magnetic flux density of the proposed coupling structure. When one of the planar receiving coils deviates from the center of the transmitting coil, the magnetic flux density decreases; however, the magnetic flux density of the plane where the other planar coil of the dual receiving coil is located increases. The total sum of magnetic flux density in two planes remains stable.

**Table 1.** The parameters of the proposed magnetic coupler.

| Parameters | Definitions | Value |
|---|---|---|
| $L_1$ | Self-inductance of transmitting coil | 104.49 µH |
| $L_2$ | Self-inductance of receiving coil | 63.58 µH |
| $a$ | Outer radius of transmitting coil | 260 mm |
| $l$ | Length of transmitting coil | 51.2 mm |
| $N_1$ | Transmitting coil turn number | 16 |
| $a_1$ | Outer radius of receiving coil | 205.6 mm |
| $a_2$ | Inner radius of receiving coil | 180 mm |
| $N_2$ | Receiving coil turn number | 8 |

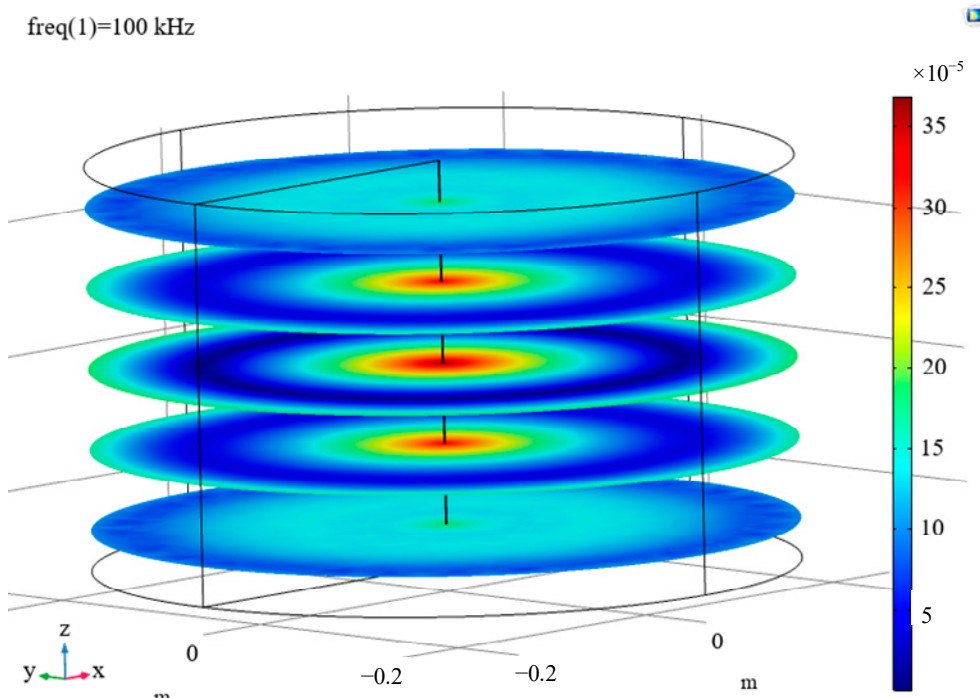

**Figure 6.** Section diagram of the magnetic flux density of the proposed coupling structure.

Figure 7 shows the configuration of the magnetic coupler; the solenoid transmitting coil is wound in the charging station, and the receiving coil is composed of planar receiving coil 1 and receiving coil 2. Each of the coils in the dual combined planar receiving coil has the same dimensions. In Figure 7, *l* represents the length of the transmitting coil, *d* represents the distance between the two receiving coils, and the axial misalignment *x* stands for the space between the center of the transmitting coil and the receiving coils.

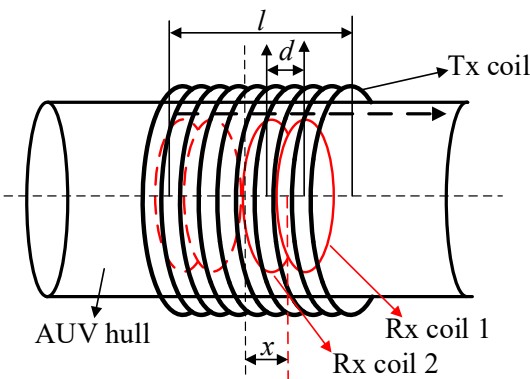

**Figure 7.** The configuration of the magnetic coupler.

In the proposed coupling structure, $M_{11}$ stands for the mutual inductance between the transmitting coil and receiving coil 1, $M_{12}$ is the mutual inductance between the transmitting coil and receiving coil 2, and $M_3$ represents the mutual inductance between receiving coil 1 and receiving coil 2. $M_{total}$ is the mutual inductance between the transmitting coil and the receiving coil. $M_{total} = M_{11} + M_{12} - M_3$. The normalized mutual inductances varying with axial misalignment are shown in Figure 8.

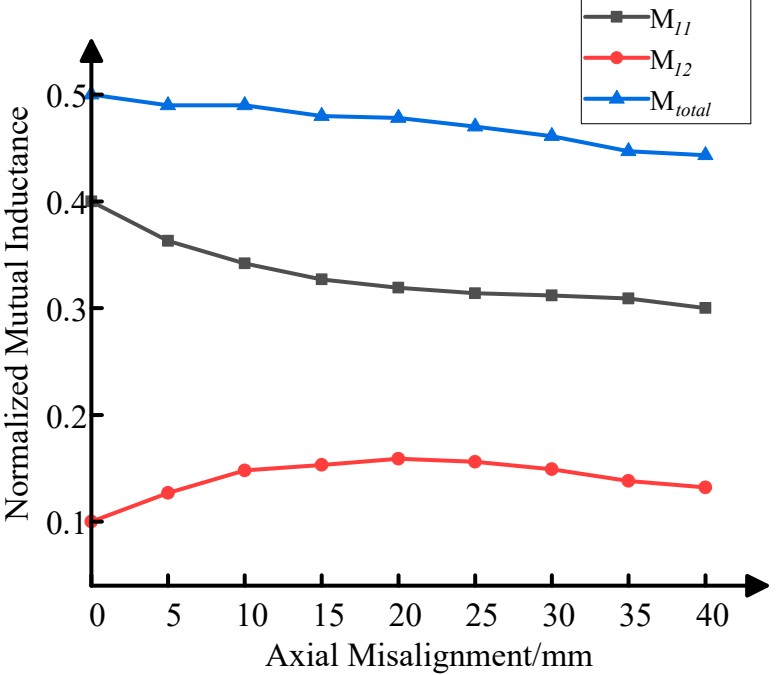

**Figure 8.** Normalized mutual inductances varying with axial misalignment.

From Figure 8, it can be observed that the mutual inductance between the transmitting coil and receiving coil 1 $M_{11}$ decreases monotonically, and the mutual inductance between the transmitting coil and receiving coil 2 $M_{12}$ first increases and then decreases with increasing axial misalignment. The mutual inductance between the transmitting coil and the receiving coil $M_{total}$ is approximately invariant.

The mutual inductance between the transmitting coil and receiving coil 1 $M_{11}$, the mutual inductance between the transmitting coil and receiving coil 2 $M_{12}$, and the mutual inductance between receiving coil 1 and receiving coil 2 $M_3$ are influenced by the distance between the two receiving coils $d$ when the axial misalignment changes. Therefore, the distance between the coils in the dual receiving coil influences the mutual inductance of the proposed magnetic coupler. Figure 9 shows the mutual inductance of the proposed magnetic coupler with different distances between the coils in the dual receiving coil versus axial misalignment.

While changing the axial misalignment changes from 0 to 40 mm, the influence of changes in the distance between the dual combined planar coils on the mutual inductance of the proposed magnetic coupler is investigated using the parametric sweep analysis method; the distances between the dual combined planar coils are varied from 1 to 20 mm. The mutual inductance of the proposed magnetic coupler with different distances between the coils in the dual receiving coil ($d$ = 5, 10, 15, 20 mm) is shown in Figure 9. It can be seen from Figure 9 that when the distance between the dual combined planar coils is 10 mm, the coupling structure has a higher mutual inductance with increasing axial misalignment. The mutual inductance changes smoothly with minimal fluctuations when the axial misalignment is increased from 0 to 40 mm. The mutual inductance reaches a maximum value of 39.4 μH, and it only decreases by 1.7%, even at the maximum offset

distance of 40 mm. Thus, the optimal distance between the dual combined planar coils is selected as 10 mm.

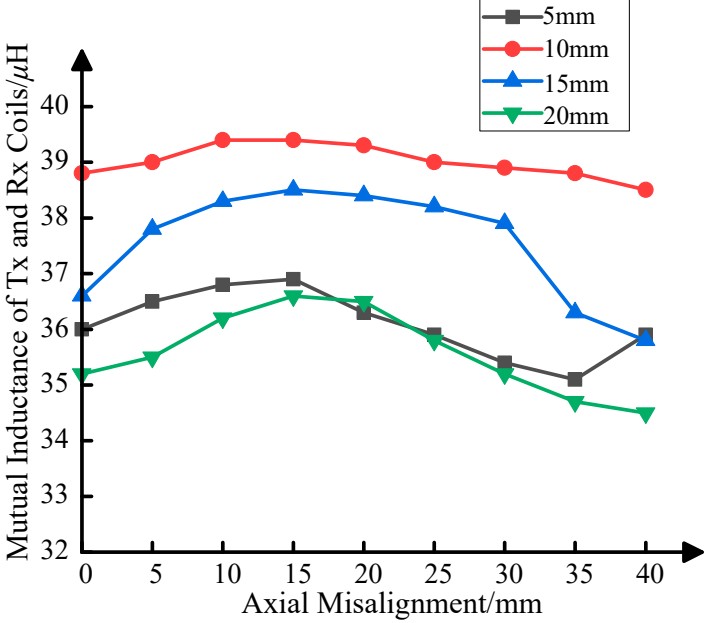

**Figure 9.** The mutual inductance of the proposed magnetic coupler with different distances between the coils in the dual receiving coil versus axial misalignment.

## 4. Experimental Verification

A WPT prototype is set up to verify the output characteristic of the presented magnetic coupler, as shown in Figure 10. AWG 38 litz wires with 400 strands are utilized to wind the magnetic coils. We use electronic load to emulate the battery, the load of which changes during the charging process. The parameters of the transmitting coil and the receiving coil are measured using the LCR meter; Table 2 presents the parameters of the WPT system.

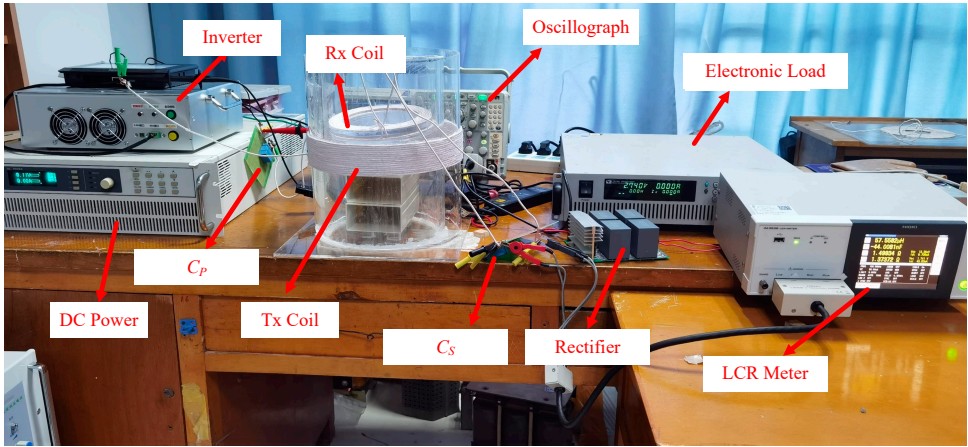

**Figure 10.** Experimental prototype.

The variation in the mutual inductance of the proposed magnetic coupler with changes in axial offset distance is shown in Figure 11. When the axial offset distance is 0, the maximum mutual inductance is 25.745 μH. In comparison with the solenoid coil to unipolar planar coil coupling structure, the mutual inductance of the proposed magnetic coupler (solenoid coil to dual combined planar coil) is increased by double. When the axial misalignment is increased from 0 to 40 mm, the mutual inductance of the proposed magnetic

coupler with the dual combined planar coil decreases more slowly compared to with the unipolar coil. When the maximum axial offset is 40 mm, the mutual inductance is 20.8 µH.

**Table 2.** The parameters of the WPT system.

| Parameters | Definitions | Value |
|---|---|---|
| $L_1$ | Self-inductance of the transmitting coil | 106.34 µH |
| $L_2$ | Self-inductance of the receiving coil | 56.07 µH |
| $R_1$ | Internal resistance of the transmitting coil | 1203.1 mΩ |
| $R_2$ | Internal resistance of the receiving coil | 565.3 mΩ |
| $f$ | System frequency | 100 kHz |
| $C_P$ | Compensation capacitor of the primary side | 23.82 nF |
| $C_S$ | Compensation capacitor of the secondary side | 45.17 nF |

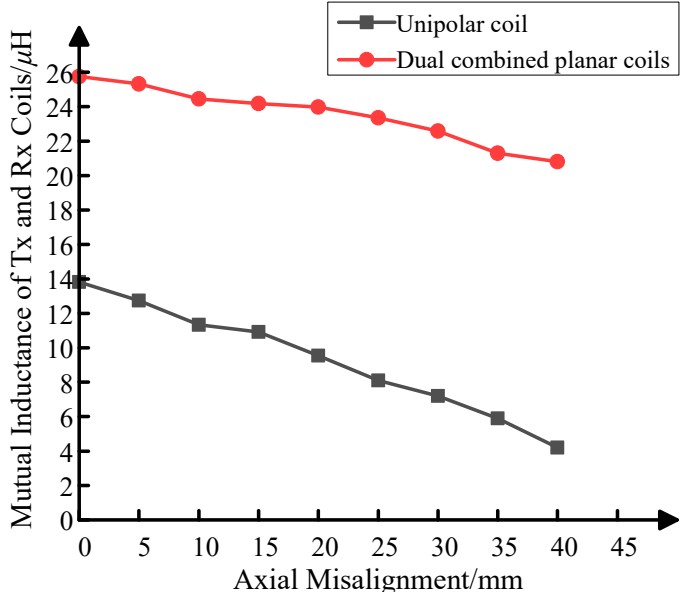

**Figure 11.** The mutual inductance of the proposed magnetic coupler varies with axial offset distance.

Figure 12 shows the waveforms of the input/output voltage and current of the proposed WPT system. On the primary side, the input voltage keeps the same phase with the input current, and on the secondary side, the output voltage is in phase with the output current. This indicates that both the primary side and the secondary side are in a resonant state.

The variation in the power transmission efficiency of the WPT system with the proposed coupling structure with the load resistance is presented in Figure 13. As is demonstrated in Figure 13, the power transmission efficiency increases from 69.9% to the maximum value of 85.04% as the load resistance increases from 10 Ω to 45 Ω; the power transmission efficiency then decreases when the load resistance increases from 45 Ω to 70 Ω. The experimental results are close to those of the theoretical analysis. Therefore, the optimal load resistance for the proposed WPT system is 45 Ω.

In order to emulate the seawater environment, saltwater with a salinity of 4‰ is employed to fill the gap between the cylindrical hulls. Figure 14 shows the power transmission efficiency of the WPT system versus axial misalignment; a comparison test of the solenoid-unipolar planar coil coupling structure is added to validate the output characteristics of the proposed magnetic coupler.

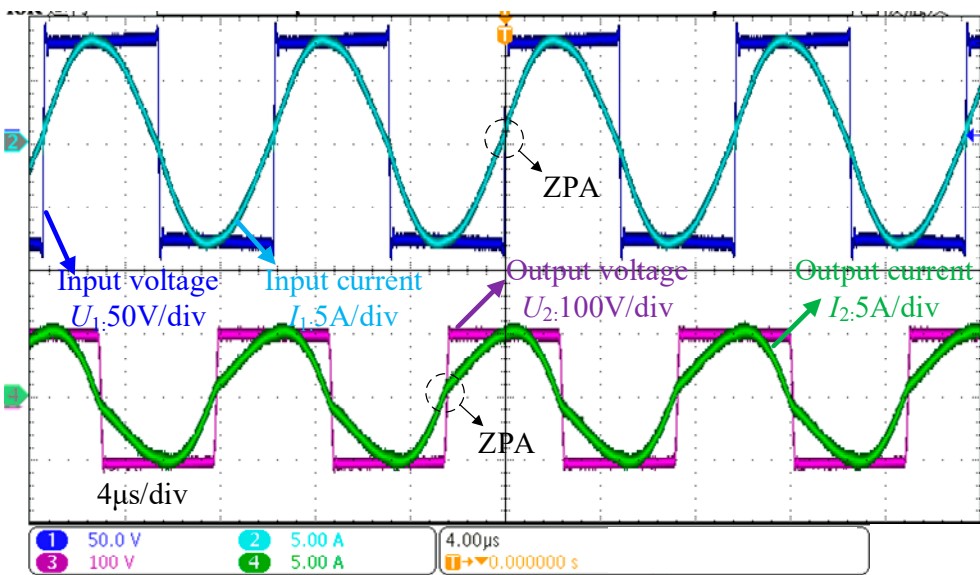

**Figure 12.** Waveforms of the input/output voltage and current of the WPT system.

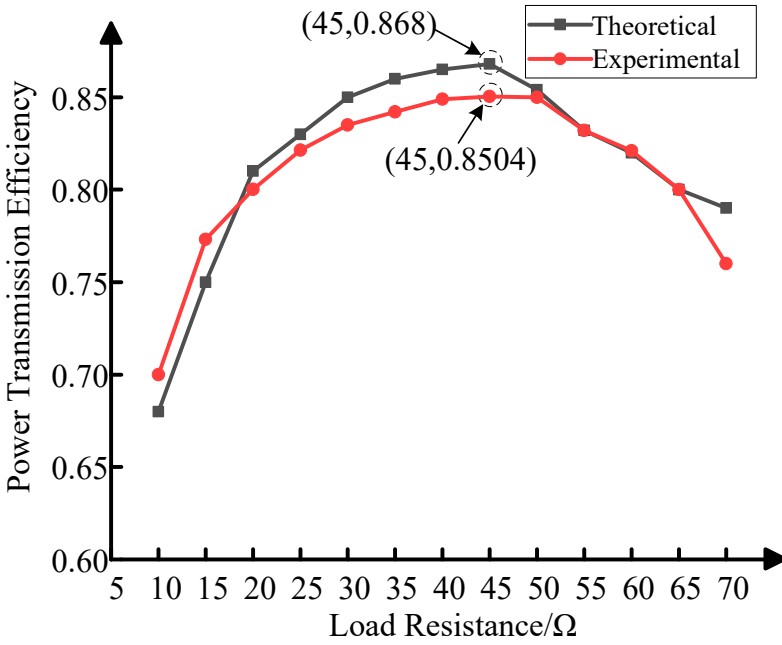

**Figure 13.** Variation in power transmission efficiency with load resistance.

As can be seen from Figure 14, the power transmission efficiency of the magnetic coupling structure with the dual combined planar coil in air is higher than the coupling structure with the unipolar coil. When the axial misalignment is increased from 0 to 40 mm, the power transmission efficiency of the WPT system with the proposed dual combined planar coil decreases from 91.64% to 86.1%, but for the WPT system with the solenoid-unipolar planar coil coupling structure, the power transmission efficiency drops from 84% to 75.2%. This indicates that the introduction of the dual combined planar receiving coil causes the power transmission efficiency of WPT system to slowly decrease with increasing axial misalignment. The proposed coupling structure has good tolerance to axial misalignment.

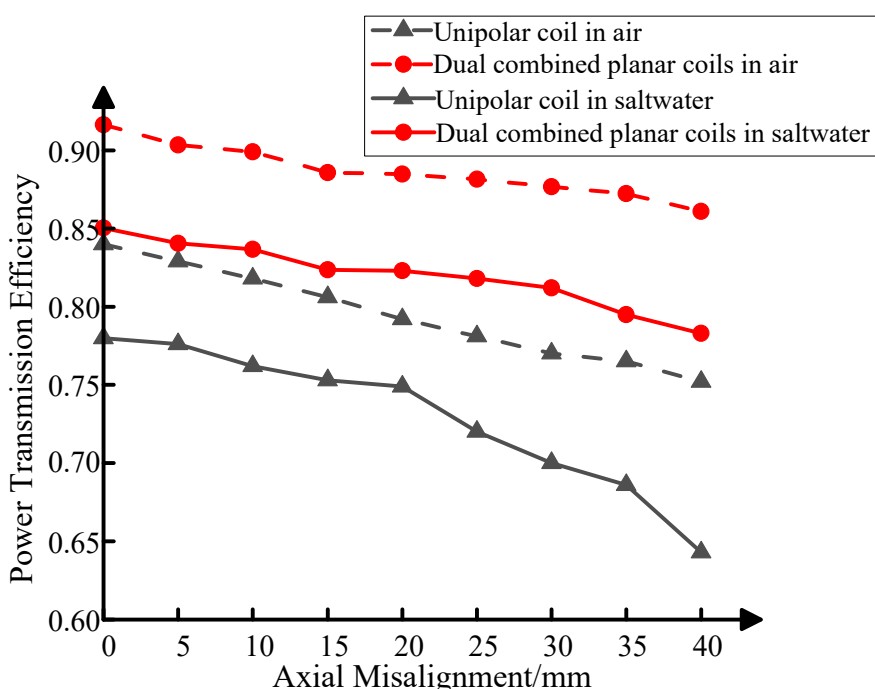

**Figure 14.** Variation in power transmission efficiency with axial misalignment.

The variation in the power transmission efficiency of the WPT system with the dual combined planar coil and the unipolar planar coil with axial misalignment in saltwater is also shown in Figure 14. As is demonstrated in Figure 14, when the axial misalignment increases from 0 to 40 mm, the power transmission efficiency of the WPT system with the unipolar planar coil in saltwater decreases from 78% to 64.3%; in contrast, the power transmission efficiency of the WPT system with the proposed dual combined planar receiving coil drops from 85.04% to 78.3%. The power transmission efficiency the of WPT system with the dual combined planar coil in saltwater decreases to a lesser extent than the WPT system with the unipolar planar coil with increasing axial misalignment, which is similar to the experimental results obtained in air. By comparing the experimental results in air and saltwater, it can be found that the power transmission efficiency of the WPT system in saltwater is lower than that in air. The high-frequency ac current in magnetic coils generates an alternating magnetic field in saltwater, and the time-varying magnetic field produces a vortex electric field in saltwater. Saltwater is a medium with good conductivity; the electromotive force would be induced, generating eddy current, which results in eddy current loss in saltwater and a decline in power transmission efficiency [26].

As is demonstrated in Figure 15, the output power increases when the axial misalignment rises from 0 to 40 mm. The secondary current increases due to the reduction in mutual inductance when axial misalignment occurs, thus increasing the output power. The output power of the magnetic coupling structure with the dual combined planar coil is higher than that of the coupling structure with the unipolar coil in air. When the axial misalignment increases from 0 to 40 mm, the output power of the WPT system with the proposed dual combined planar coil rises from 436.28 W to 449.06 W, representing a variation in output power of 2.93%, while for the WPT system with solenoid-unipolar planar coil coupling structure, the output power increases from 375.89 W to 410.21 W, representing a variation in output power of 9.13%. This indicates that the stability of the WPT system's output characteristics is enhanced by the proposed coupling structure when axial misalignment occurs.

Figure 15 shows the output power of the WPT system versus axial misalignment. In the experiment, the input voltage is set to 80 V.

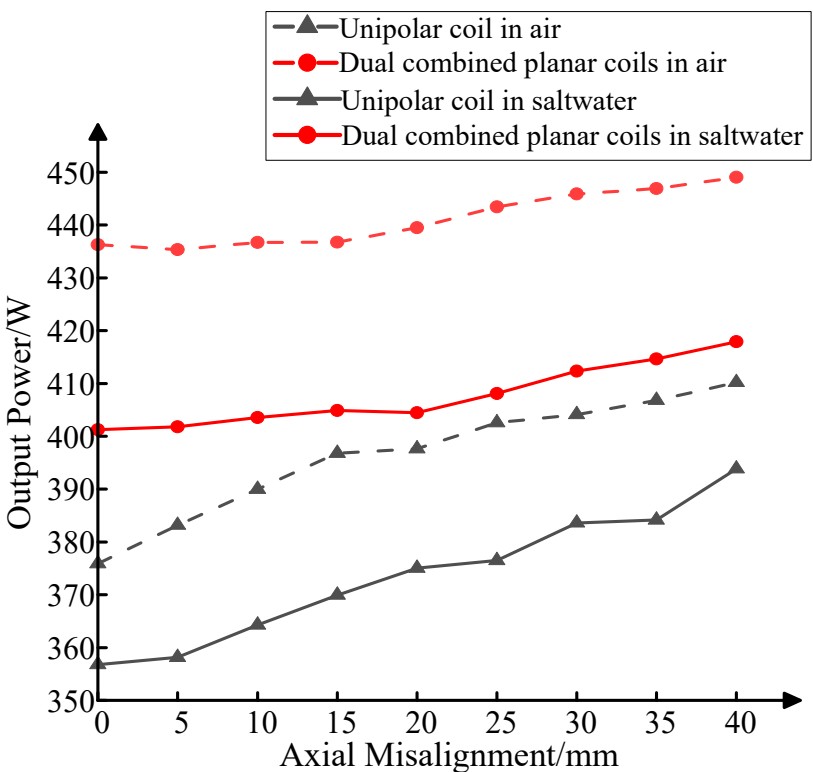

**Figure 15.** The output power versus axial misalignment.

The variation of the output power of the WPT system with a dual combined planar coil and a unipolar planar coil with axial misalignment in saltwater is also shown in Figure 15. When the axial misalignment increases from 0 to 40 mm, the output power of the WPT system with the unipolar planar coil in saltwater increases from 356.76 W to 393.84 W, representing a variation in output power of 10.4%. In contrast, the power transmission efficiency of the WPT system with the proposed dual combined planar receiving coil rises from 401.23 W to 417.89 W, and the deviation rate of output power is 4.15%. The output power of the WPT system with the dual combined planar coil in saltwater exhibits less variation than the WPT system with the unipolar planar coil with increasing axial misalignment, which is similar to the experimental results obtained in air.

## 5. Discussion

In this paper, a magnetic coupling structure is proposed that consists of a solenoid transmitting coil and dual combined planar receiving coil for a wireless charging system for AUVs. The main function of this novel coupling structure is to enhance the misalignment tolerance of the WPT system, especially in the axial direction under the dynamic marine environment. The mutual inductance between the solenoid transmitting coil and the dual combined planar receiving coil remains nearly constant when misalignment changes within the limit.

However, the proposed solenoid-dual combined planar magnetic coupling structure does not generate a uniform magnetic field, as misalignment varies. In future work, the magnetic coupling structure will be greatly improved by adding magnetic cores or optimizing the parameters of the magnetic coils, so that the coupling structure generates a uniform magnetic field; thereby, both the stability of the power transmission efficiency and the output power of the system could be further enhanced. Furthermore, the introduction of the dual combined planar receiving coil increases the cost of winding compared with the unipolar receiving coil, which should be taken into account in the economic analysis of WPT system.

## 6. Conclusions

A WPT system with a novel coupling structure for AUVs was put forward in this article. The magnetic coupling structure was composed of a solenoid as the transmitting coil and dual combined planar coils as the receiving coil. Both the theoretical analysis and the experimental results show that the presented coupling structure can significantly eliminate deviations in the mutual inductance under different axial misalignments compared with solenoid-unipolar planar coil coupling structure. It was validated by experiment that the proposed magnetic coupler effectively improved the power transmission efficiency and output power stability within the possible axial misalignments. When the axial misalignment was increased from 0 to 40 mm, the power transmission efficiency of WPT system with the proposed dual combined planar receiving coil dropped from 85.04% to 78.3% in saltwater, and the variation in output power was lower than 4.15%; both values are smaller than when using the coupling structure with the unipolar coil.

**Author Contributions:** Conceptualization, writing—original draft preparation, H.W. and P.W.; methodology, writing—review and editing, H.W., P.W. and J.L.; software, P.W. and J.L.; validation, data curation, P.W., J.L. and J.Y.; formal analysis, J.Y. and J.L.; investigation, H.W. and L.Y.; resources, L.Y. and X.T.; visualization, J.Y. and Y.Z.; supervision, K.Z. and X.T.; project administration, funding acquisition, Y.Z., K.Z. and H.W. All authors have read and agreed to the published version of the manuscript.

**Funding:** This work was supported by the National Natural Science Foundation of China under Grant 52171338, the China Postdoctoral Science Foundation under Grant 2021M702638 and 2023MD734218, and the Natural Science Basic Research Plan in Shaanxi Province of China under Grant 2023-JC-QN-0475, and the Starting Research Fund for Postdoctoral Researcher from Xi'an University of Technology under Grant 119-451121005.

**Institutional Review Board Statement:** Not applicable.

**Informed Consent Statement:** Not applicable.

**Data Availability Statement:** Not applicable.

**Conflicts of Interest:** The authors declare no conflict of interest.

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
