# Peer review of "Improving the Misalignment Tolerance of Wireless Power Transfer System for AUV with Solenoid-Dual Combined Planar Magnetic Coupler"

_jmse, doi:10.3390/jmse11081571_

Round 1

Reviewer 1 Report

Overall, the introduction provides a clear overview of the problem statement and the importance of addressing the power transfer model in the AUV domain. However, there are a few areas that could be improved for clarity and precision. Here are my comments:

a)      The starting letter of the abbreviations "WPT, AUV, …." need to be like Wireless Power Transfer (WPT) instead of wireless power transfer.

b)     When introducing related work, it is important to focus on the literature related to WPT and AUV. However, the paper only mentions a few related literature, with other literature having relatively low relevance. In addition, the specific contributions or findings are not highlighted. Also, you need to investigate works like in " https://doi.org/10.3390/s19214762" and compare it with your solution from different points of view.

c)      Contributions are not provided in the introduction section. It would be helpful to briefly mention the contributions here or indicate that they will be discussed in detail later.

d)     Paper organization has not been described clearly.

2. The description of the network model and its components is generally clear, but it would be beneficial to provide more specific explanations in underwater environments. Clarifying the research problems addressed by the model, the functionalities of the AUV.

3. There are obvious errors in formula 14; There is a clear conflict between formula 8 and formula 15;

4. It would be beneficial to provide more insights into the advantages and potential challenges of the proposed model. Additionally, further discussion on the practical implications would enhance the clarity and completeness of the proposed model.

5. There are some problems, such as Grammar, formatting, and other issues.

Minor

Author Response

First, we would like to thank you for your kind letter and for reviewers' constructive comments concerning our article. These comments are all valuable and helpful for improving our article. All the authors have seriously discussed about all these comments. According to the reviewers' comments, we have modified our manuscript to meet with the requirements of your journal. In this revised version, changes to our manuscript within the document are highlighted by using red-colored text. Point-by-point responses to the reviewers are listed below in this letter.

Reviewer 1:

Comments and Suggestions for Authors

Overall, the introduction provides a clear overview of the problem statement and the importance of addressing the power transfer model in the AUV domain. However, there are a few areas that could be improved for clarity and precision. Here are my comments:

Response:

Thank you very much for your time involved in reviewing the manuscript. We appreciate your clear and detailed feedback and hope that the explanation has fully addressed all of your concerns. In this letter, we discuss each of your comments individually. To facilitate this discussion, we first restate your comments in italic font, and our responses are provided directly afterward in red text.

Comment 1 a):

-  The starting letter of the abbreviations "WPT, AUV, …." need to be like Wireless Power Transfer (WPT) instead of wireless power transfer.

Response 1 a):

Thanks very much for your comments. We have checked the abbreviations in our article throughly, and modifications have been made in the manuscript, all changes to our manuscript are highlighted by using red-colored text.

Comment 1 b):

-  When introducing related work, it is important to focus on the literature related to WPT and AUV. However, the paper only mentions a few related literature, with other literature having relatively low relevance. In addition, the specific contributions or findings are not highlighted. Also, you need to investigate works like in " https://doi.org/10.3390/s19214762" and compare it with your solution from different points of view.

Response 1 b):

Thank you for the great suggestion. We have reviewed all the references in our article, according to your advice, some related work have been added to the introduction and the related literature have been added to references, such as references [1], [2], [10], [11], [20], [22]. Also, some relatively low relevance literature have been deleted. The specific contributions and findings have been highlighted in the manuscript.

  1. Avilash, S.; Santosha, K.D.; Dwivedy ; Robi, P.S. Advancements in the field of autonomous underwater vehicle. Ocean Engineering 2019, 181, 145-160.
  2. Muhammad, F.; Rizwan, A.B.; Basit, R.; Hani, A.; Muhammad, W.A.; Syed, B.S.; Md, A.N.; Vehbi, C.G. A Cross-Layer QoS Channel-Aware Routing Protocol for the Internet of Underwater Acoustic Sensor Networks. Sensors 2019, 19, 4762.( https://doi.org/10.3390/s19214762)
  3. Yang, C.; Lin, M.; Li, D. Improving Steady and Starting Characteristics of Wireless Charging for an AUV Docking System. IEEE Jounal of Oceanic Engineering 2020, 45, 430-441.
  4. Orekan, T.; Zhang, P.; Shih, C. Analysis, Design, and Maximum Power-Efficiency Tracking for Undersea Wireless Power Transfer. IEEE Journal of Emerging and Selected Topics in Power Electronics 2018, 6, 843-854.
  5. Yan, Z.; Song, B.; Zhang, Y.; Zhang, K.; Mao, Z.; Hu, Y. A rotation-free wireless power transfer system with stable output power and efficiency for autonomous underwater vehicles. IEEE Transactions on Power Electronics 2019, 34, 4005-4008.
  6. Zhang, K.; Zhang, X.; Zhu Z.; Yan, Z.; Song, B.; Mi, C.C. A new coil structure to reduce eddy current loss of WPT systems for underwater vehicles. IEEE Transactions on Vehicular Technology. 2019, 68, 245-253.

The following literature has been deleted.

Tan, P.; Xu, W.; Shangguan, X.; Wu, Y.; Liu, H. Mutual Inductance Modeling and Parameter Optimization of Wireless Power Transfer System with Combined Series-Wound Hexagonal Coils. Transactions of China Electrotechnical Society 2023, 38, 2299-2309.

Comment 1 c):

-  Contributions are not provided in the introduction section. It would be helpful to briefly mention the contributions here or indicate that they will be discussed in detail later.

Response 1 c):

Thanks very much for your comments. We have briefly provided the main contributions of our work in the introduction part. It is indicated in the paragraph about paper organization that the contributions will be discussed in detail in Section 5.

Comment 1 d):

-  Paper organization has not been described clearly.

Response 1 d):

Thank you for pointing this out. We have added a paragraph about the organization of this article at the end of introduction part. It is highlighted by using red-colored text in the manuscript.

Comment 2:

-  The description of the network model and its components is generally clear, but it would be beneficial to provide more specific explanations in underwater environments. Clarifying the research problems addressed by the model, the functionalities of the AUV.

Response 2:

Thanks very much for your suggestion. According to your advice, we have added some specific explanations about the underwater environments, the research problems addressed by the model, the functionalities of the AUV in the introduction part. They are highlighted by using red-colored text.

Comment 3:

-  There are obvious errors in formula 14; There is a clear conflict between formula 8 and formula 15;

Response 3:

We were really sorry for our careless mistakes, thanks for your kind reminder. Formula 14, formula 8 and 15 have been checked and modified.

Comment 4:

-  It would be beneficial to provide more insights into the advantages and potential challenges of the proposed model. Additionally, further discussion on the practical implications would enhance the clarity and completeness of the proposed model.

Response 4:

Thank you for the great suggestion on improving our manuscript. According to your suggestion, we have added the discussion part in Section 5 to discuss the advantages, potential challenges and practical implications of the proposed magnetic coupling structure in detail.

Comment 5:

-  There are some problems, such as Grammar, formatting, and other issues.

Response 5:

Thanks for your kind reminder, and we were really sorry for our careless mistakes. We have checked the English grammar, formatting and writing skills of our article throughly, and modifications have been made in the manuscript, all changes are highlighted by using red-colored text.

Reviewer 2 Report

The paper discusses a very interesting topic. The technical work sounds well. A minor comment as follows:

1. Please highlight the main contribution of the paper at the end of the introduction.

2. What is the main issue of using the proposed method? Please briefly highlight it somewhere in your discussion.

Overall is good and satisfactory

Author Response

First, we would like to thank you for your kind letter and for reviewers' constructive comments concerning our article. These comments are all valuable and helpful for improving our article. All the authors have seriously discussed about all these comments. According to the reviewers' comments, we have modified our manuscript to meet with the requirements of your journal. In this revised version, changes to our manuscript within the document are highlighted by using red-colored text. Point-by-point responses to the reviewers are listed below in this letter.

Reviewer 2:

Comments to the Author

The paper discusses a very interesting topic. The technical work sounds well. A minor comment as follows:

Response:

Thank you very much for your time involved in reviewing the manuscript. We appreciate your clear and detailed feedback and hope that the explanation has fully addressed all of your concerns. In this letter, we discuss each of your comments individually. To facilitate this discussion, we first restate your comments in italic font, and our responses are provided directly afterward in red text.

Comment 1:

- Please highlight the main contribution of the paper at the end of the introduction.

Response 1:

Thank you for the great suggestion. We have provided the main contributions of the paper in the introduction part. Also, the contributions, advantages and potential challenges of the proposed magnetic coupling structure are discussed in detail in Section 5. It is mentioned at the end of the introduction. They are highlighted by using red-colored text in the manuscript.

Comment 2:

- What is the main issue of using the proposed method? Please briefly highlight it somewhere in your discussion.

Response 2:

Thanks very much for your comments on improving our manuscript. According to your suggestion, we have added the discussion part in Section 5 to discuss the advantages, potential challenges and main issue of using the proposed magnetic coupling structure in detail.

Reviewer 3 Report

The authors proposed a novel magnetic coupling structure with improved misalignment tolerance of WPT system for AUV. The theoretical analysis seems solid as well as the experimental validation. Please elaborate the result in Fig.9 why the optimal distance between the dual combined planar coils is 10 mm. The economic analysis is needed to mention for using dual combined planar coils compared with unipolar coil. The authors can also elaborate the the poor performance of the operation in saltwater medium in the theoretical analysis in order to support the experimental result.

English seems fine, but the manuscript arrangement can be improved. For example, the full stop in the line 148 of page 5 should have a space to the next sentence. Most of the subscript of parameters are inconsistent. For example, transmitting coil current uses Ip (instead of primary coil current) and receiving coil current uses Is (instead of secondary coil current) as well as sometimes uses small letter case in subscription but in equation (4) and (5) the capital letter case is used. The italic style is also used in subscription as well. 

Author Response

First, we would like to thank you for your kind letter and for reviewers' constructive comments concerning our article. These comments are all valuable and helpful for improving our article. All the authors have seriously discussed about all these comments. According to the reviewers' comments, we have modified our manuscript to meet with the requirements of your journal. In this revised version, changes to our manuscript within the document are highlighted by using red-colored text. Point-by-point responses to the reviewers are listed below in this letter.

Reviewer 3:

Comments to the Author

- The authors proposed a novel magnetic coupling structure with improved misalignment tolerance of WPT system for AUV. The theoretical analysis seems solid as well as the experimental validation.

Response:

Thank you very much for your time involved in reviewing the manuscript. We appreciate your clear and detailed feedback and hope that the explanation has fully addressed all of your concerns. In this letter, we discuss each of your comments individually. To facilitate this discussion, we first restate your comments in italic font, and our responses are provided directly afterward in red text.

Comment 1:

- Please elaborate the result in Fig.9 why the optimal distance between the dual combined planar coils is 10 mm.

Response 1:

Thanks very much for your comments. According to your advice, we have added explanation in detail about how to determine the optimal distance between the dual combined planar coils before and after Figure 9. It is highlighted by using red-colored text in the manuscript.

Comment 2:

- The economic analysis is needed to mention for using dual combined planar coils compared with unipolar coil.

Response 2:

Thank you for pointing this out. We have added the discussion part in Section 5 to discuss the advantages, potential challenges and main issue of using the proposed magnetic coupling structure in detail. The economic analysis is mentioned when using dual combined planar coils compared with unipolar coil in WPT system for AUV.

Comment 3:

- The authors can also elaborate the poor performance of the operation in saltwater medium in the theoretical analysis in order to support the experimental result.

Response 3:

Thank you for the great suggestion. The poor performance of the operation in saltwater medium is caused by eddy current loss, as saltwater is a medium with good conductivity. According to your suggestion, we have added explanation and reference in detail to support the experimental result.

Comment 4:

-English seems fine, but the manuscript arrangement can be improved. For example, the full stop in the line 148 of page 5 should have a space to the next sentence. Most of the subscript of parameters are inconsistent. For example, transmitting coil current uses Ip (instead of primary coil current) and receiving coil current uses Is (instead of secondary coil current) as well as sometimes uses small letter case in subscription but in equation (4) and (5) the capital letter case is used. The italic style is also used in subscription as well.

Response 4:

Thanks very much for your comments. We have checked the arrangement of our article throughly, and modifications have been made in the manuscript, including some mistakes in formatting. The  subscript of parameters are unified in the manuscript. All changes to our manuscript are highlighted by using red-colored text.
